# Association of Cardiovascular Risk Factors between Adolescents and Their Parents Is Mitigated by Parental Physical Activity—A Cross-Sectional Study

**DOI:** 10.3390/ijerph192114026

**Published:** 2022-10-28

**Authors:** William R. Tebar, Gerson Ferrari, Jorge Mota, Ewerton P. Antunes, Beatriz A. S. Aguilar, Javier Brazo-Sayavera, Diego G. D. Christofaro

**Affiliations:** 1Post-Graduation Program in Movement Sciences, Faculty of Sciences and Technology, São Paulo State University (Unesp), Presidente Prudente 19060-900, Brazil; 2Center of Clinical and Epidemiological Research, University Hospital, University of Sao Paulo, Sao Paulo 05508-000, Brazil; 3Faculty of Health Sciences, Universidad Autónoma de Chile, Providencia 7500912, Chile; 4Research Center on Physical Activity, Health and Leisure (CIAFEL), Faculty of Sport, University of Porto, 4200-450 Porto, Portugal; 5Department of Sports and Computer Science, Universidad Pablo de Olavide, 41013 Seville, Spain

**Keywords:** parent–child, hypertension, substance use, overweight, exercise

## Abstract

Introduction: It is hypothesized that children’s habits can be modulated by their parent’s lifestyle. However, it is still not established whether the relationship between cardiovascular risk factors (CVRF) in adolescents and their parents could be attenuated by parental physical activity levels. Objective: This study aimed to analyze the relationship of CVRF between adolescents and their parents according to parental physical activity level. Methods: A school-based sample of 1231 adolescents, 1202 mothers and 871 fathers were included (*n* = 3304). The CVRF assessed were overweight, hypertension, smoking and alcohol consumption. The parental physical activity level was assessed using a validated questionnaire, being classified into physically active and inactive parents. The statistical analysis considered all parents and stratification by physical activity level. Results: The prevalence of CVRF was higher in fathers than in mothers (70.6% vs. 54.9% for overweight, 23.3% vs. 19.7% for hypertension, 17.9% vs. 12.4% for smoking and 60.4% vs. 28.5% for alcohol consumption). Adolescents with active mothers showed lower prevalence of overweight (13.9% vs. 19.6%), while adolescents with active fathers showed higher prevalence of alcohol consumption (23.5% vs. 16.9%). The CVRF of both fathers and mothers were positively associated with CVRF of adolescents. However, the association of CVRF between adolescents and their parents was mitigated among active parents, while all the CVRF remained associated in physically inactive parents. Conclusion: The parental physical activity level seems to mitigate the association of CVRF between adolescents and their parents. The promotion of an active lifestyle at the family level can contribute to reduce CVRF among adolescents.

## 1. Introduction

Youth is an important period of life in which lifestyle habits are likely to be carried to further life stages, and a high prevalence of risk factors in adolescents is observed [1,2,3]. It was reported that parental behaviors, such as smoking, alcohol consumption and low physical activity level, were associated with higher cardiovascular risk factors in their children [4].

Yang et al. [5] in a study with children and adolescents in China, found that young people with overweight parents were more likely to have metabolic syndrome. Vázquez-Rodríguez et al. [6], in a study with Mexican adolescents, observed a relationship between smoking in parents and their children. Similar findings were observed for the relationship between high alcohol consumption by parents and adolescents [7]. However, adolescents whose parents have healthy lifestyle habits are more likely to replicate this type of behavior. A study conducted with Brazilian adolescents observed that those who had active mothers and fathers were more likely to be more physically active [8]. Schoeppe et al. [9] observed that active sports and commuting by mothers were associated with greater sports practice and active commuting by their children.

Furthermore, it was reported in the literature that physical activity practice is inversely related to obesity, arterial hypertension, smoking and alcohol consumption [10,11,12,13] both in adult and adolescents [14,15,16,17]. Based on this premise, it is hypothesized that the association of cardiovascular risk factors between adolescents and their parents could be mitigated if they were physically active, when compared to less active parents. Therefore, the aim of the present study was to analyze the association between cardiovascular risk factors (overweight, high blood pressure, smoking and alcohol consumption) in adolescents and their parents, according to the parental level of physical activity.

## 2. Material and Methods

### 2.1. Sample

The sample was comprised of adolescents, fathers and mothers from the city of Londrina, located in the southern region of Brazil, with a population of approximately 550,000 inhabitants. The adolescents were recruited in a school-based sample process. The director of each selected school was contacted and informed about the research objectives. Afterwards, the researchers explained the research objectives to the students in the classrooms, inviting the adolescents to participate. Those adolescents who agreed to participate received a parental consent form authorizing the participation in the study and two questionnaires about lifestyle habits for their parents if they also agreed to participate.

In this study, the six largest schools in the central region of Londrina were selected, as they received students from all regions of the city. To calculate the minimum sample size, it was considered a prevalence of 50% of outcome (used in epidemiological studies with several outcomes or unknown prevalence) [18], a tolerable error of 4% and a sampling power of 80%. Due to the present study using cluster sampling (schools and classrooms), a design effect correction of 1.5 was applied, which increased the number of participants by 50%. Finally, 20% were added for avoiding loss of power due to sample losses, so thefinal sample size needed was 1044 adolescents.

This research project was approved by the Ethics and Research Committee of Universidade Estadual de Londrina. Parents and/or guardians signed a consent form authorizing young people to participate in the study.

### 2.2. Parents Physical Activty

Parental habitual physical activity was assessed by the questionnaire proposed by Baecke et al. [19]. This instrument comprised three different domains and was previously validated for the Brazilian population [20,21]. This instrument assessed the domains of occupational physical activities, sports practice and other types of physical activities at leisure-time and commuting. The Baecke’s questionnaire provided a dimensionless score for each assessed domain, and the sum of these scores provided the total physical activity score. For the purpose of the present study, the physical activity score was stratified into quartiles, where parents in the highest quartile were classified as physically active and those in the lower quartiles were classified as physically inactive.

### 2.3. Overweight

The participants’ overweight was defined though the body mass index by dividing the body weight in kilograms by the height squared in meter. The body mass of the adolescents was measured by a digital scale and the height was measured using a portable wall-fixed stadiometer. The classification of overweight in adolescents was >+1SD cutoffs from specific values proposed by World Health Organization [22], according to age and sex. Among parents, the values of body mass and height were self-reported and used to calculate body mass index, where parents with body mass index of 25 kg/m² or higher were classified as overweight.

### 2.4. Hypertension

The definition of hypertension among adolescents was considered by objective measurements of systolic and diastolic blood pressure by a digital oscillometric device (OMRON HEM-742), which was previously validated for this population [23]. Appropriately sized cuffs were used for the arms of young people following the recommendations in the literature [24]. Measurements were taken on the right arm, with the participant in seated position and with a minimum rest of five minutes before the first measurement and with a 2-min rest interval between the first and the second measurements. The mean of the two measurements was calculated, and the cutoff points recommended by the National High Blood Pressure Education Program (NHBPEP) were adopted [25]. Adolescents with blood pressure values above the 95th percentile for their respective age, height and gender were considered to have hypertension.

The parents were asked whether they had a medical diagnosis of hypertension or were using blood pressure lowering medications. Those parents who answered “yes” for at least one of these conditions were classified as having hypertension.

### 2.5. Smoking

The smoking habit was measured by two specific questions, in which the adolescent and parents answered whether they currently smoked and the number of cigarettes they smoked per day. Those participants who reported they currently smoked at least one cigarette were considered to be smokers.

### 2.6. Alcohol Consumption

The Brazilian translated and adapted version of the Substance Abuse and Mental Health Services Administration (SAMHSA) questionnaire from the U.S. Department of Health and Human Services [26] was used to assess the alcohol consumption of the sample. This instrument asks how often the participant drinks alcoholic beverages (beer, ice drinks, wine, and distilled: vodka, whisky, brandy, rum). The responses were accounted for each type of beverage (I do not drink, 1–2 days/month, 3–4 days/month, 1–2 days/week, 3–4 days/week, 5–6 days a week, every day), and approximately how many doses were consumed (responses were: none, 1–2 doses a day, 3–4 doses a day, 5–6 doses a day, 7–10 doses a day, more than 10 doses a day). One dose was defined according to 14 g of alcohol (standard dose), being 350 mL for beer, 275 mL for ice drinks, 150 mL for wine, and 45 mL for distilled beverages. Adolescents and parents who reported to drink alcohol at a frequency equal to or greater than 1–2 days a week and with 1–2 drinks a day were considered to have high alcohol consumption.

### 2.7. Socioeconomic Level

The Brazilian Criteria for Economic Classification (ABEP) [27] was used to assess the socioeconomic status of the sample. This instrument considers the level of education of the head of family, the presence of a housemaid and the presence and quantity of specific rooms and consumer goods at home and provides a specific score that classifies the households into economic classes from highest to lowest: A, B1, B2, C1, C2, D–E, being further classified into high (A, B1), middle (B2, C1) and low (C2, D–E) socioeconomic classes.

### 2.8. Statistics Analysis

The sample characterization variables were presented as mean and standard deviation for continuous data and in frequency for categorical data. The proportions of cardiovascular risk factors according to the level of physical activity were compared by chi-square test. The association of the cardiovascular risk factors of the parents and their children was analyzed by binary logistic regression, adjusted for the age (parents and adolescents), for the adolescents’ sex, for parental education level and by family’s socioeconomic level. The statistical significance was defined as *p* < 0.05 and the confidence level adopted was 95%. The statistical package used was SPSS version 25.0.

## 3. Results

A final sample of 1231 adolescents was assessed, 52.8% being females (*n* = 716). From these adolescents, a total of 2073 parents were assessed, comprised of 1202 mothers (58%) and 871 fathers (42%). Overall, this study included data from 3304 participants. Sample characteristics are presented in Table 1. A higher prevalence of cardiovascular risk factors among fathers was observed, the prevalence of alcohol consumption being twice as high than in mothers.

Figure 1 presents the prevalence of cardiovascular risk factors in parents and adolescents according to the parental level of physical activity. It was observed that inactive fathers had a higher prevalence of hypertension when compared to active fathers (Panel A). When the prevalence of cardiovascular risk factors in adolescents was analyzed according to the physical activity level of their parents, it was observed that adolescents with inactive mothers had a higher prevalence of overweight, whereas adolescents with active fathers had a higher prevalence of alcohol consumption (Panel B).

Table 2 presents the association of cardiovascular risk factors in adolescents with cardiovascular risk factors in their mothers. In the model considering all mothers, the presence of cardiovascular risk factor of mothers was positively associated with cardiovascular risk factors in adolescents. In the analysis stratified by mothers’ level of physical activity, all the cardiovascular risk factors remained associated with physically inactive mothers, while only overweight remained associated with physically active mothers.

The association between cardiovascular risk factors in adolescents and their fathers is presented in Table 3. When considering all fathers, a positive association with overweight, hypertension and alcohol consumption was observed between adolescents and their fathers. In physically inactive fathers, all the cardiovascular risk factors remained associated between adolescents and fathers, whereas only hypertension and alcohol consumption remained associated in physically active fathers.

## 4. Discussion

The results of the current study show that cardiovascular risk factors in adolescents were positively associated with cardiovascular risk factors in their parents. When stratified by parental physical activity level, it was observed that cardiovascular risk factors in adolescents and their parents were strongly associated with physically inactive in comparison to physically active parents.

The association of overweight between adolescents and parents in the present study corroborated with a previous study by Lassere et al. [28], which observed that overweight in parents was associated with overweight in their children regardless of confounding factors. Some reasons for the observed findings may be linked to genetic factors [29], but also to obesogenic lifestyle habits of the adolescents’ parents [30]. However, this association was not observed when considering the overweight of physically active fathers and their children. Sigmund et al. [31] in a study with 1101 parent–child dyads observed that when the father was more active at leisure and had more than 10,000 steps a day, these behaviors significantly reduced the odds of obesity in their children. These findings are related with previous results, where it was observed that children of active fathers would be more likely to be more active [32], which could contribute to the reduction in the chances of being at risk of overweight. In addition, it is possible that active parents provided better support for the practice of physical activity of their children [33].

Overweight/obesity in adolescents is an important problem from the perspective of social development since it is associated with social discrimination and risk of depression [34], body image dissatisfaction [35] and limitations for professional insertion and affective relationships [36]. This is a worrying condition once a recent meta-analysis reported that prevalence of overweight and obese in Brazilian adolescents is high and continues to rise in the last decades without sex differences, being 8.2% before the year 2000, 18.9% from 2000 to 2009 and 25.1% from 2010 forward [37]. These factors highlight the need for interventions that encompass the different determinants of obesity in youth and present the important role of the family environment for this purpose.

In addition, the economic situation of the family is an important covariate of the present study. Previous findings reported that parental socioeconomic status can predict the physical activity level of children [38]. The economic condition significantly affected the food intake of parents and of adolescents [39], where the cost of foods and familiar income exert influence in the eating choices [40]. Additionally, it was observed that people with higher SES may have higher alcohol consumption than those with lower SES, but otherwise, this last group can be more susceptible to negative alcohol-related consequences [41]. However, the present study results remained significant even after adjustment by the socioeconomic status of the family.

Considering the association of hypertension between adolescents and their parents, it was observed that this association lost significance in physically active mothers. Corroborating the findings of the present study, Ranasinghe et al. [42], in a study considering the history of hypertension among parents, grandparents, siblings and children, observed that participants who had a history of hypertension were more likely to be physically inactive. One of the possible reasons is that physical activity can modify the association between genetic load and hypertension [43].

Smoking in adolescents and their parents was associated only between those who were physically inactive. Andrade et al. [44], in a study with a Brazilian sample, observed that adolescents whose mothers were smokers were twice as likely to be smokers. Oztekin et al. [45] observed that the smoking status of the mother was associated with smoking at an early age in boys. One of the possible reasons for these findings is that physical activity practice is inversely related to smoking. Salin et al. [46], in a cohort study, found that in the trajectory from childhood to midlife, participants who were persistently active were less likely to be smokers in adulthood. Previous studies have shown that the mothers’ physical activity practice is associated with the practice of physical activity by their children [8,32] which could contribute to this non-association between children of physically active mothers. The association between smoking in children of inactive mothers with their mothers’ smoking habits could occur due to the greater freedom of experimenting with cigarettes replicated in this type of habit in their mothers.

Regarding the association of alcohol consumption between adolescents and their parents, it was observed that this association lost significance in physically active mothers. On the other hand, the alcohol consumption of adolescents and their fathers remained associated regardless of the physical activity level of fathers. Our findings are partially corroborated by the results of Kabwama et al. [7], in which study an association between alcohol consumption in Ugandan adolescents and youth and maternal alcohol consumption was observed, but such associations were not observed with paternal consumption. One of the reasons for the discrepancies between studies may be due to the fact that Kabwama’s study had only male adolescents and young adults in its sample and a variation in the age of the sample. A positive association of alcohol consumption with high levels of physical activity was observed in Brazilian young, middle-aged and older adults [47]. Such cultural issues could influence the observed results, since alcohol consumption is present in socialization time after sports practice [48].

Upon the findings of the present study and the current evidence in the literature, the family environment plays a fundamental role in adolescent health. It was observed in a Brazilian sample of 842 adolescents and both their parents that healthy eating habits in parents were associated with healthy eating habits in their children, as well as the opposite: unhealthy eating habits in parents and their children were associated [49]. The family environment can also contribute to mental health in Brazilian adolescents, where living with parents and encouragement of a healthy lifestyle presented a lower chance of having common mental disorders [50]. These arguments highlight the need for health promotion strategies in youth to be focused on the family environment. A harmful situation for further actions is related to the COVID-19 pandemic consequences on Brazilian adult lifestyle behaviors, since an increase in the frequency of smoking, fast-food consumption and alcohol drinking was observed, in addition to a decrease in physical activity levels and the consumption of fruits and vegetables [51]. This behavioral change may substantially affect the family environment and impair the introduction of positive habits in the household, being an important burden for adolescent health. 

As limitations of this study, it is important to highlight the cross-sectional design, which precluded cause and effect relationships. Another aspect is that the physical activity practice was evaluated only through a questionnaire, which was susceptible to memory and classification bias. However, the difficulty in using accelerometers in studies carried out in developing countries with a large number of participants is noteworthy. Furthermore, to the best of our knowledge, this was the first study to consider the association of cardiovascular risk factors between adolescents and their parents according to the parental physical activity level.

## 5. Conclusions

In conclusion, the findings of the present study indicate that cardiovascular risk factors were more associated between adolescents and physically inactive parents than among adolescents whose parents were physically active. Therefore, health promotion actions, especially considering the practice of physical activity, should include the family environment as a whole.

## Figures and Tables

**Figure 1 ijerph-19-14026-f001:**
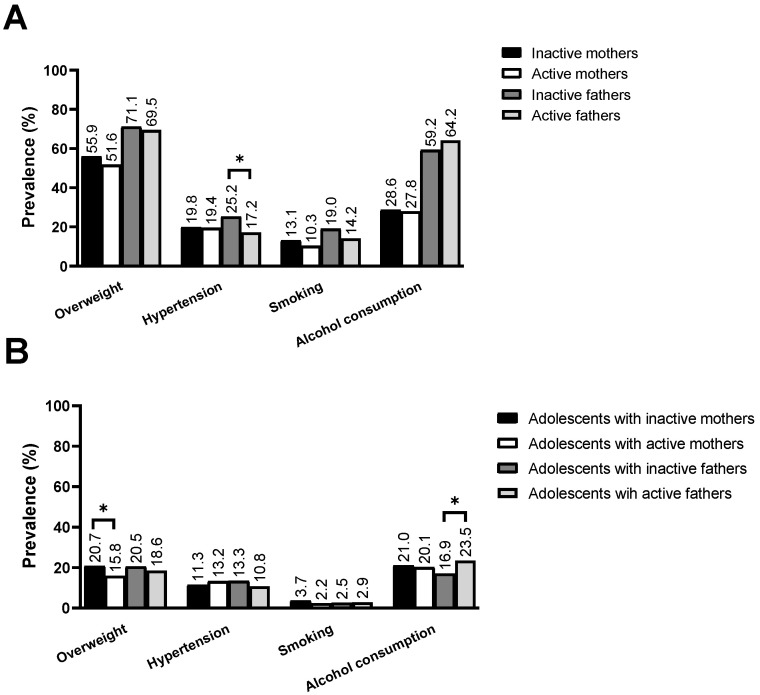
Prevalence of cardiovascular risk factors in parents according to their level of physical activity. * *p* < 0.05 for comparison of proportions by chi-square test.

**Table 1 ijerph-19-14026-t001:** Sample characterization.

	Adolescents (*n* = 1231)	Mothers (*n* = 1202)	Fathers (*n* = 871)
Age (years), mean (SD)	15.6 (1.1)	43.3 (7.2)	45.8 (7.5)
Overweight, *n* (%)	246 (20.0)	660 (54.9)	615 (70.6)
Hypertension, *n* (%)	148 (12.0)	237 (19.7)	203 (23.3)
Smoking, *n* (%)	42 (3.4)	149 (12.4)	156 (17.9)
Alcohol consumption, *n* (%)	255 (20.7)	342 (28.5)	526 (60.4)

**Table 2 ijerph-19-14026-t002:** Association between the presence of cardiovascular risk factors in mothers and in their children according to the physical activity level of mothers.

All mothers (*n* = 1202)—Odds ratio (95% confidence interval)				
**Adolescents CVRF**	**Overweight**	**Hypertension**	**Smoking**	**Alcohol Consumption**
Overweight	3.67 (2.63–5.14), *p* < 0.001	---	---	---
Hypertension	---	2.63 (1.75–3.94), *p* < 0.001	---	---
Smoking	---	---	2.35 (1.10–4.99), *p* = 0.027	---
Alcohol consumption	---	---	---	1.99 (1.47–2.69), *p* < 0.001
Physically inactive mothers (*n* = 929)—Odds ratio (95% confidence interval)				
**Adolescents CVRF**	**Overweight**	**Hypertension**	**Smoking**	**Alcohol Consumption**
Overweight	3.77 (2.59–5.50), *p* < 0.001	---	---	---
Hypertension	---	3.21 (2.01–5.12), *p* < 0.001	---	---
Smoking	---	---	3.02 (1.38–6.62), *p* = 0.006	---
Alcohol consumption	---	---	---	2.19 (1.56–3.07), *p* < 0.001
Physically active mothers (*n* = 273)—Odds ratio (95% confidence interval)				
**Adolescents CVRF**	**Overweight**	**Hypertension**	**Smoking**	**Alcohol Consumption**
Overweight	3.36 (1.59–7.11), *p* = 0.001	---	---	---
Hypertension	---	1.52 (0.64–3.62), *p* = 0.341	---	---
Smoking	---	---	NCO	---
Alcohol consumption	---	---	---	1.42 (0.72–2.81), *p* = 0.309

Adjusted by age (adolescents and mothers), mothers’ educational level and by adolescents’ sex and socioeconomic level. CVRF = Cardiovascular risk factors; NCO = No cases observed.

**Table 3 ijerph-19-14026-t003:** Association between the presence of cardiovascular risk factors in fathers and in their children according to the physical activity level of fathers.

All fathers (*n* = 871)—Odds ratio (95% confidence interval)				
**Adolescents CVRF**	**Overweight**	**Hypertension**	**Smoking**	**Alcohol Consumption**
Overweight	1.76 (1.17–2.63), *p* = 0.006	---	---	---
Hypertension	---	2.14 (1.38–3.33), *p* = 0.001	---	---
Smoking	---	---	2.37 (0.95–5.91), *p* = 0.064	---
Alcohol consumption	---	---	---	1.92 (1.30–2.83), *p* = 0.001
Physically inactive fathers (*n* = 667)—Odds ratio (95% confidence interval)				
**Adolescents CVRF**	**Overweight**	**Hypertension**	**Smoking**	**Alcohol Consumption**
Overweight	2.09 (1.30–3.37), *p* = 0.002	---	---	---
Hypertension	---	1.81 (1.11–2.96), *p* = 0.018	---	---
Smoking	---	---	3.21 (1.12–9.17), *p* = 0.030	---
Alcohol consumption	---	---	---	1.83 (1.16–2.89), *p* = 0.009
Physically active fathers (*n* = 204)—Odds ratio (95% confidence interval)				
**Adolescents CVRF**	**Overweight**	**Hypertension**	**Smoking**	**Alcohol Consumption**
Overweight	1.08 (0.48–2.42), *p* = 0.856	---	---	---
Hypertension	---	4.95 (1.67–14.7), *p* = 0.004	---	---
Smoking	---	---	1.20 (0.13–11.3), *p* = 0.876	---
Alcohol consumption	---	---	---	2.14 (1.01–4.54), *p* = 0.047

Adjusted by adolescents’ and fathers’ age, adolescents’ sex, socioeconomic level, and fathers’ education level. CVRF = Cardiovascular risk factors.

## Data Availability

The data presented in this study are available upon reasonable request from the corresponding author. The data are not publicly available due to o ethical commitments for sensitive patient information.

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
