# Peer review of "Association of Cardiovascular Risk Factors between Adolescents and Their Parents Is Mitigated by Parental Physical Activity—A Cross-Sectional Study"

_ijerph, 2022, doi:10.3390/ijerph192114026_

Round 1
Reviewer 1 Report
I have two main methodological remarks, if corrected, the article can be published.
1. For adolescents, it is customary to indicate not the body mass index, but SDS BMI. It would be expedient, despite small age variations, to recalculate these values, given that the main results of the article are based on these data.
2. When describing Alcohol consumption, the authors use the questionnaire of the Brazilian Center for Information on Psychotropic Drugs [26], the content of which is difficult for me to assess. It would be advisable to clarify in more detail which doses of alcohol and how are considered in the questionnaire: light and strong alcohol and the volume of the dose in each option.
Author Response
Dear Editor and Reviewers,
We would like to thank for the opportunity to resubmit the attached manuscript entitled “Association of cardiovascular risk factors between adolescents and their parents is mitigated by parental physical activity – a cross-sectional study” (ijerph-1967354) for possible publication in the International Journal of Environmental Research and Public Health.
The authors also wish to express their gratitude for the efforts of the editor and reviewers in directing the manuscript towards a more acceptable form for publication. The manuscript has been carefully checked, and appropriate changes made following the reviewers’ suggestions.
All the revisions in the manuscript were marked using the “Track Changes” function of MS Word and we have attached a separate document containing point-by-point responses to the reviewer comments.
The authors hope that the added revisions adequately address the comments and we are available for any other queries.
Sincerely,
The authors
Reviewer 1
I have two main methodological remarks, if corrected, the article can be published.
- For adolescents, it is customary to indicate not the body mass index, but SDS BMI. It would be expedient, despite small age variations, to recalculate these values, given that the main results of the article are based on these data.
Response: Dear reviewer, we thank for the comment. We changed the previous definition of adolescents overweight from Cole et al. (2000) to the cutoffs of >+1SD for age and sex as proposed by World Health Organization (WHO, 2007). Upon these changes, the prevalence of adolescent overweight changed from 18.6 to 20.0 (please see Table 1), as well as the proportion of overweight adolescents according to physical activity of parents also changed (please see Figure 1). The odds ratio from logistic regression models regarding adolescent overweight were also updated according to new analysis. However, significant associations remained the same (please see Tables 2 and 3).
Reference:
World Health Organization. Growth reference charts 2-5 and 5-19 years. Retrieved from: http://www.who.int/childgrowth/standards/bmi_for_age/en/ Accessed 19 October 2022
- When describing Alcohol consumption, the authors use the questionnaire of the Brazilian Center for Information on Psychotropic Drugs [26], the content of which is difficult for me to assess. It would be advisable to clarify in more detail which doses of alcohol and how are considered in the questionnaire: light and strong alcohol and the volume of the dose in each option.
Response: We thank for the comment. The information about alcohol consumption was clarified in the Methods, please see below or at lines from 124 to 137 of revised version with track changes.
“The Brazilian translated and adapted version of SAMHSA (Substance Abuse and Mental Health Services Administration) questionnaire from U.S. Department of Health and Human Services [26] was used to assess the alcohol consumption of the sample. This instrument asks about how often the participant drink alcoholic beverages (bier, ice drinks, wine, and distilled: vodka, whisky, brandy, rum). The responses were accounted for each type of beverage (I don't drink, 1-2 days/month, 3-4 days/month, 1-2 days/week, 3-4 days/week, 5-6 days a week, all the days), and about how many doses have been consumed (responses were: none, 1-2 doses a day, 3-4 doses a day, 5-6 doses a day, 7-10 doses a day, more than 10 doses a day). One dose was de-fined according to 14g of alcohol (standard dose), being 350ml for bier, 275ml for ice drinks, 150ml for wine, and 45ml for distilled beverages. Adolescents and parents who reported to drink alcohol at a frequency equal to or greater than 1-2 days a week and with 1-2 drinks a day were considered to have high alcohol consumption.”
Reference:
Galduróz et al. [Use of psychotropic drugs in Brazil: household survey in the 107 biggest Brazilian cities--2001]. Revista Latino-Americana de Enfermagem. 2005;13: 888-95. doi:10.1590/s0104-11692005000700017

Reviewer 2 Report
I congratulate the authors on their choice of research topic. The issues raised in the article are very interesting and arouse the interest of the reviewer.
Parents are the most important people in the life of children, they are with them from the first moments of life, they look after them and spend most of their time with them. It is from their parents that children learn the world and similarly acquire eating habits. Parents play a huge role here because children learn through imitation. Therefore, the daily choices and attitudes of parents influence the child's eating behavior and physical activity.
In my opinion:
• A clear introduction is used in the manuscript;
• The material and research methods are described in detail;
• Research has been properly planned;
• Appropriate analytical methods were selected for the analysis of the results;
• The results of the analyzes are presented in graphical and tabular form.
I am asking for an answer to my doubts:
1. Can the economic situation of the family have an impact on poor eating habits, drinking alcohol and low physical activity?
2. What percentage of children in Brazil are overweight and obese? Is it a significant problem from the perspective of social development?
3. Please explain in a few sentences how parents' approach to introducing positive habits in households has changed in recent years in Brazil.
Author Response
Dear Editor and Reviewers,
We would like to thank for the opportunity to resubmit the attached manuscript entitled “Association of cardiovascular risk factors between adolescents and their parents is mitigated by parental physical activity – a cross-sectional study” (ijerph-1967354) for possible publication in the International Journal of Environmental Research and Public Health.
The authors also wish to express their gratitude for the efforts of the editor and reviewers in directing the manuscript towards a more acceptable form for publication. The manuscript has been carefully checked, and appropriate changes made following the reviewers’ suggestions.
All the revisions in the manuscript were marked using the “Track Changes” function of MS Word and we have attached a separate document containing point-by-point responses to the reviewer comments.
The authors hope that the added revisions adequately address the comments and we are available for any other queries.
Sincerely,
The authors
Reviewer 2
I congratulate the authors on their choice of research topic. The issues raised in the article are very interesting and arouse the interest of the reviewer.
Parents are the most important people in the life of children, they are with them from the first moments of life, they look after them and spend most of their time with them. It is from their parents that children learn the world and similarly acquire eating habits. Parents play a huge role here because children learn through imitation. Therefore, the daily choices and attitudes of parents influence the child's eating behavior and physical activity.
In my opinion:
- A clear introduction is used in the manuscript;
- The material and research methods are described in detail;
- Research has been properly planned;
- Appropriate analytical methods were selected for the analysis of the results;
- The results of the analyzes are presented in graphical and tabular form.
I am asking for an answer to my doubts:
- Can the economic situation of the family have an impact on poor eating habits, drinking alcohol and low physical activity?
Response: We thank the reviewer for the valuable comments. Previous findings reported that parental SES can predict the physical activity level of children (Mutz et al., 2017). It has been previously observed that the correlation between food intake of parents and of adolescents depends on socioeconomic level (Da Veiga et al., 2006), since the cost of foods and familiar income exert influence in the eating choices (Lo et al., 2009). About the association between socioeconomic status and alcohol consumption, it has been reported that people with higher SES may have higher alcohol consumption than those with lower SES, but otherwise this last group can be more susceptible to negative alcohol-related consequences (Collings, 2016). In our sample, we did not observed association between overweight and socioeconomic status of adolescents, but fathers with higher socioeconomic status showed higher prevalence of physically actives, while no association was observed among mothers. However, our results remained significant even when adjusted by the family SES. We have addressed these arguments at Discussion section in a specific paragraph. Please see at lines from 221 to 229 of revised version with track changes.
References:
Mutz M, Albrecht P. Parents' Social Status and Children's Daily Physical Activity: The Role of Familial Socialization and Support. J Child Fam Stud. 2017;26(11):3026-3035. doi:10.1007/s10826-017-0808-3
Da Veiga GV, Sichieri R. Correlation in food intake between parents and adolescents depends on socioeconomic level. Nutr Res. 2006;26:517–523.
Lo, Yuan-Ting et al. “Health and nutrition economics: diet costs are associated with diet quality.” Asia Pacific journal of clinical nutrition vol. 18,4 (2009): 598-604.
Collins SE. Associations Between Socioeconomic Factors and Alcohol Outcomes. Alcohol Res. 2016;38(1):83-94.
- What percentage of children in Brazil are overweight and obese? Is it a significant problem from the perspective of social development?
Response: Dear reviewer, a recent meta-analysis by Sbaraini et al. (2021) reported that prevalence of overweight and obese in Brazilian adolescents is high and continue to rise in the last decades without sex differences, being 8.2% until year 2000, 18.9% from 2000 to 2009, and 25.1% in 2010 forward. Through the prism of social development, we believe that overweight/obesity has an important burden, since obesity has been associated with social discrimination and risk of depression (Puhl et al., 2010), body image dissatisfaction (Tebar et al., 2020) and limitations for professional insertion and affective relationships (Reinehr, 2018). These factors highlight the need for interventions which encompass the different determinants of obesity at youth and presented the important role of family environment for this purpose. We considered these arguments at Discussion section, please see at lines from 212 to 220 of revised version with track changes.
References:
Puhl RM, Heuer CA. Obesity stigma: important considerations for public health. Am J Public Health. 2010;100(6):1019-1028.
Reinehr, Thomas. “Long-term effects of adolescent obesity: time to act.” Nature reviews. Endocrinology vol. 14,3 (2018): 183-188.
Sbaraini, Mariana et al. “Prevalence of overweight and obesity among Brazilian adolescents over time: a systematic review and meta-analysis.” Public health nutrition vol. 24,18 (2021): 6415-6426.
Tebar WR, Gil FCS, Scarabottolo CC, Codogno JS, Fernandes RA, Christofaro DGD. Body size dissatisfaction associated with dietary pattern, overweight, and physical activity in adolescents: A cross-sectional study. Nurs Health Sci. 2020;22(3):749-757.
- Please explain in a few sentences how parents' approach to introducing positive habits in households has changedin recent years in Brazil.
Response: We appreciate the comment. We considered that family environment plays a fundamental role in adolescent health. It has been observed in a Brazilian sample of 842 adolescents and their both parents that healthy eating habits of parents was associated with healthy eating habits of their children, as well as the opposite: unhealthy eating habits of parents and their children were associated (Christofaro et al., 2020). The family environment can also contribute to mental health of Brazilian adolescents, where living with parents and encouragement of a healthy lifestyle presented a lower chance of having common mental disorders (Gratão et al., 2022). These arguments highlight the need to health promotion strategies in youth be focused on the family environment. A harmful situation for further actions is related to Covid-19 pandemic consequences on Brazilian adult lifestyle behaviors, since has been observed an increase in the frequency of smoking, fast-food consumption and alcohol drinking, besides a decrease in physical activity level and consumption of fruits and vegetables (Souza et al., 2022). This behavioral change may substantially affect family environment and impair the introduction of positive habits at household, being an important burden for adolescent health. A specific paragraph was inserted at Discussion encompassing these arguments for the parental influence in the introduction of positive habits at households, please see at lines from 263 to 279 of revised version with track changes.
References:
Christofaro, Diego G D et al. “Gender Analyses of Brazilian Parental Eating and Activity With Their Adolescents' Eating Habits.” Journal of nutrition education and behavior vol. 52,5 (2020): 503-511.
Gratão LHA, Pessoa MC, Rocha LL, et al. Living with parents, lifestyle pattern and common mental disorders in adolescents: a school-based study in Brazil. BMC Public Health. 2022;22(1):1862.
Souza TC, Oliveira LA, Daniel MM, et al. Lifestyle and eating habits before and during COVID-19 quarantine in Brazil. Public Health Nutr. 2022;25(1):65-75.

Round 2
Reviewer 1 Report
I am satisfied with the corrections made by the authorsI am satisfied with the corrections made by the authors.